# Data-based analysis, modelling and forecasting of the COVID-19 outbreak

**Cleo Anastassopoulou[1]\*, Lucia Russo[2], Athanasios Tsakris[1], Constantinos Siettos[ID][3]\***

**1** Department of Microbiology, Medical School, University of Athens, Athens, Greece, **2** Consiglio Nazionale delle Ricerche, Science and Technology for Energy and Sustainable Mobility, Napoli, Italy, **3** Dipartimento di Matematica e Applicazioni "Renato Caccioppoli", Università degli Studi di Napoli Federico II, Napoli, Italy

\* constantinos.siettos@unina.it (CS); cleoa@med.uoa.gr (CA)

## Abstract

Since the first suspected case of coronavirus disease-2019 (COVID-19) on December 1st, 2019, in Wuhan, Hubei Province, China, a total of 40,235 confirmed cases and 909 deaths have been reported in China up to February 10, 2020, evoking fear locally and internationally. Here, based on the publicly available epidemiological data for Hubei, China from January 11 to February 10, 2020, we provide estimates of the main epidemiological parameters. In particular, we provide an estimation of the case fatality and case recovery ratios, along with their 90% confidence intervals as the outbreak evolves. On the basis of a Susceptible-Infectious-Recovered-Dead (SIDR) model, we provide estimations of the basic reproduction number ($R_0$), and the per day infection mortality and recovery rates. By calibrating the parameters of the SIRD model to the reported data, we also attempt to forecast the evolution of the outbreak at the epicenter three weeks ahead, i.e. until February 29. As the number of infected individuals, especially of those with asymptomatic or mild courses, is suspected to be much higher than the official numbers, which can be considered only as a subset of the actual numbers of infected and recovered cases in the total population, we have repeated the calculations under a second scenario that considers twenty times the number of confirmed infected cases and forty times the number of recovered, leaving the number of deaths unchanged. Based on the reported data, the expected value of $R_0$ as computed considering the period from the 11th of January until the 18th of January, using the official counts of confirmed cases was found to be ∼4.6, while the one computed under the second scenario was found to be ∼3.2. Thus, based on the SIRD simulations, the estimated average value of $R_0$ was found to be ∼2.6 based on confirmed cases and ∼2 based on the second scenario. Our forecasting flashes a note of caution for the presently unfolding outbreak in China. Based on the official counts for confirmed cases, the simulations suggest that the cumulative number of infected could reach 180,000 (with a lower bound of 45,000) by February 29. Regarding the number of deaths, simulations forecast that on the basis of the up to the 10th of February reported data, the death toll might exceed 2,700 (as a lower bound) by February 29. Our analysis further reveals a significant decline of the case fatality ratio from January 26 to which various factors may have contributed, such as the severe control measures taken in Hubei, China (e.g. quarantine and hospitalization of infected individuals), but mainly because of the fact that the actual cumulative numbers of infected and recovered cases in

**Data Availability Statement:** The data used in this paper were acquired from https://gisanddata.maps.arcgis.com/apps/opsdashboard/index.html#/bda7594740fd40299423467b48e9ecf6. In S1 Table we provide the data that we have used for

this study, i.e. the cumulative confirmed cases of infected recovered and deaths from January 11 to February 10.

**Funding:** The authors received no specific funding for this work.

**Competing interests:** The authors have declared that no competing interests exist.

the population most likely are much higher than the reported ones. Thus, in a scenario where we have taken twenty times the confirmed number of infected and forty times the confirmed number of recovered cases, the case fatality ratio is around ~0.15% in the total population. Importantly, based on this scenario, simulations suggest a slow down of the outbreak in Hubei at the end of February.

## Introduction

An outbreak of "pneumonia of unknown etiology" in Wuhan, Hubei Province, China in early December 2019 has spiraled into an epidemic that is ravaging China and threatening to reach a pandemic state [1]. The causative agent soon proved to be a new betacoronavirus related to the Middle East Respiratory Syndrome virus (MERS-CoV) and the Severe Acute Respiratory Syndrome virus (SARS-CoV). The novel coronavirus SARS-CoV-2 disease has been named "COVID-19" by the World Health Organization (WHO) and on January 30, the COVID-19 outbreak was declared to constitute a Public Health Emergency of International Concern by the WHO Director-General [2]. Despite the lockdown of Wuhan and the suspension of all public transport, flights and trains on January 23, a total of 40,235 confirmed cases, including 6,484 (16.1%) with severe illness, and 909 deaths (2.2%) had been reported in China by the National Health Commission up to February 10, 2020; meanwhile, 319 cases and one death were reported outside of China, in 24 countries [3].

The origin of COVID-19 has not yet been determined although preliminary investigations are suggestive of a zoonotic, possibly of bat, origin [4, 5]. Similarly to SARS-CoV and MERS-CoV, the novel virus is transmitted from person to person principally by respiratory droplets, causing such symptoms as fever, cough, and shortness of breath after a period believed to range from 2 to 14 days following infection, according to the Centers for Disease Control and Prevention (CDC) [1, 6, 7]. Preliminary data suggest that older males with comorbidities may be at higher risk for severe illness from COVID-19 [6, 8, 9]. However, the precise virologic and epidemiologic characteristics, including transmissibility and mortality, of this third zoonotic human coronavirus are still unknown.

Using the serial intervals (SI) of the two other well-known coronavirus diseases, MERS and SARS, as approximations for the true unknown SI, Zhao et al. estimated the mean basic reproduction number ($R_0$) of SARS-CoV-2 to range between 2.24 (95% CI: 1.96-2.55) and 3.58 (95% CI: 2.89-4.39) in the early phase of the outbreak [10]. Very similar estimates, 2.2 (95% CI: 1.4-3.9), were obtained for $R_0$ at the early stages of the epidemic by Imai et al. 2.6 (95% CI: 1.5-3.5) [11], as well as by Li et al., who also reported a doubling in size every 7.4 days [1]. Wu et al. estimated the $R_0$ at 2.68 (95% CI: 2.47–2.86) with a doubling time every 6.4 days (95% CI: 5.8–7.1) and the epidemic growing exponentially in multiple major Chinese cities with a lag time behind the Wuhan outbreak of about 1–2 weeks [12].

Amidst such an important ongoing public health crisis that also has severe economic repercussions, we reverted to mathematical modelling that can shed light to essential epidemiologic parameters that determine the fate of the epidemic [13]. Here, we present the results of the analysis of time series of epidemiological data available in the public domain [14–16] (WHO, CDC, ECDC, NHC and DXY) from January 11 to February 10, 2020, and attempt a three-week forecast of the spreading dynamics of the emerged coronavirus epidemic in the epicenter in mainland China.

## Methodology

Our analysis was based on the publicly available data of the new confirmed daily cases reported for the Hubei province from the 11th of January until the 10th of February [14–16]. Based on the released data, we attempted to estimate the mean values of the main epidemiological parameters, i.e. the basic reproduction number $R_0$, the case fatality ($\hat{\gamma}$) and case recovery ($\hat{\beta}$) ratios, along with their 90% confidence intervals. However, as suggested [17], the number of infectious, and consequently the number of recovered, people is likely to be much higher. Thus, in a second scenario, we have also derived results by taking twenty times the number of reported cases for the infectious and forty times the number for the recovered cases, while keeping constant the number of deaths that is more likely to be closer to the real number. Furthermore, by calibrating the parameters of the SIRD model to fit the reported data, we also provide tentative forecasts until the 29th of February.

The basic reproduction number ($R_0$) is one of the key values that can predict whether the infectious disease will spread into a population or die out. $R_0$ represents the average number of secondary cases that result from the introduction of a single infectious case in a totally susceptible population during the infectiousness period. Based on the reported data of confirmed cases, we provide estimations of the $R_0$ from the 16th up to the 20th of January in order to satisfy as much as possible the hypothesis of $S \approx N$ that is a necessary condition for the computation of $R_0$.

We also provide estimations of the case fatality ($\hat{\gamma}$) and case recovery ($\hat{\beta}$) ratios over the entire period using a rolling window of one day from the 11th of January to the 16th of January to provide the very first estimations.

Furthermore, we calibrated the parameters of the SIRD model to fit the reported data. We first provide a coarse estimation of the recovery ($\beta$) and mortality rates ($\gamma$) of the SIRD model using the first period of the outbreak. Then, an estimation of the infection rate $\alpha$ is accomplished by "wrapping" around the SIRD simulator an optimization algorithm to fit the reported data from the 11th of January to the 10th of February. We have started our simulations with one infected person on the 16th of November, which has been suggested as a starting date of the epidemic and run the SIR model until the 10th of February. Below, we describe analytically our approach.

Let us start by denoting with $S(t)$, $I(t)$, $R(t)$, $D(t)$, the number of susceptible, infected, recovered and dead persons respectively at time $t$ in the population of size $N$. For our analysis, we assume that the total number of the population remains constant. Based on the demographic data for the province of Hubei $N = 59m$. Thus, the discrete SIRD model reads:

$$S(t) = S(t-1) - \frac{\alpha}{N} S(t-1)I(t-1) \tag{1}$$

$$I(t) = I(t-1) + \frac{\alpha}{N} S(t-1)I(t-1) - \beta I(t-1) - \gamma I(t-1) \tag{2}$$

$$R(t) = R(t-1) + \beta I(t-1) \tag{3}$$

$$D(t) = D(t-1) + \gamma I(t-1) \tag{4}$$

The above system is defined in discrete time points $t = 1, 2, \ldots$, with the corresponding initial condition at the very start of the epidemic: $S(0) = N - 1$, $I(0) = 1$, $R(0) = D(0) = 0$. Here, $\beta$ and $\gamma$ denote the "effective/apparent" per day recovery and fatality rates. Note that these parameters do not correspond to the actual per day recovery and mortality rates as the new

cases of recovered and deaths come from infected cases several days back in time. However, one can attempt to provide some coarse estimations of the "effective/apparent" values of these epidemiological parameters based on the reported confirmed cases using an assumption and approach described in the next section.

### Estimation of the basic reproduction number from the SIRD model

Let us first start with the estimation of $R_0$. Initially, when the spread of the epidemic starts, all the population is considered to be susceptible, i.e. $S \approx N$. Based on this assumption, by Eqs (2), (3) and (4), the basic reproduction number can be estimated by the parameters of the SIRD model as:

$$R_0 = \frac{\alpha}{\beta + \gamma} \tag{5}$$

Let us denote with $\Delta I(t) = I(t) - I(t-1)$, $\Delta R(t) = R(t) - R(t-1)$, $\Delta D(t) = D(t) - D(t-1)$, the reported new cases of infectious, recovered and dead at time $t$, with $C\Delta I(t)$, $C\Delta R(t)$, $C\Delta D(t)$ the cumulative numbers of confirmed cases at time $t$. Thus:

$$C\Delta X(t) = \sum_{i=1}^{t} \Delta X(t), \tag{6}$$

where, $X = I, R, D$.

Let us also denote by $\Delta \mathbf{X}(t) = [\Delta X(1), \Delta X(2), \cdots, \Delta X(t)]^T$ the $t \times 1$ column vector containing all the reported new cases up to time $t$ and by $\mathbf{C} \Delta \mathbf{X}(t) = [C\Delta X(1), C\Delta X(2), \cdots, C\Delta X(t)]^T$, the $t \times 1$ column vector containing the corresponding cumulative numbers up to time $t$. On the basis of Eqs (2), (3) and (4), one can provide a coarse estimation of the parameters $R_0$, $\beta$ and $\gamma$ as follows.

Starting with the estimation of $R_0$, we note that as the province of Hubei has a population of 59m, one can reasonably assume that for any practical means, at least at the beginning of the outbreak, $S \approx N$. By making this assumption, one can then provide an approximation of the expected value of $R_0$ using Eqs (5), (2), (3) and (4). In particular, substituting in Eq (2), the terms $\beta I(t-1)$ and $\gamma I(t-1)$ with $\Delta R(t) = R(t) - R(t-1)$ from Eq (3), and $\Delta D(t) = D(t) - D(t-1)$ from Eq (4) and bringing them into the left-hand side of Eq (2), we get:

$$I(t) - I(t-1) + R(t) - R(t-1) + D(t) - D(t-1) = \frac{\alpha}{N} S(t-1)I(t-1) \tag{7}$$

Adding Eqs (3) and (4), we get:

$$R(t) - R(t-1) + D(t) - D(t-1) = \beta I(t-1) + \gamma I(t-1) \tag{8}$$

Finally, assuming that for any practical means at the beginning of the spread that $S(t-1) \approx N$ and dividing Eq (7) by Eq (8) we get:

$$\frac{I(t) - I(t-1) + R(t) - R(t-1) + D(t) - D(t-1)}{R(t) - R(t-1) + D(t) - D(t-1)} = \frac{\alpha}{\beta + \gamma} = R_0 \tag{9}$$

Note that one can use directly Eq (9) to compute $R_0$ with regression, without the need to compute first the other parameters, i.e. $\beta$, $\gamma$ and $\alpha$.

At this point, the regression can be done either by using the differences per se, or by using the corresponding cumulative functions (instead of the differences for the calculation of $R_0$ using Eq (9)). Indeed, it is easy to prove that by summing up both sides of Eqs (7) and (8) over time and then dividing them, we get the following equivalent expression for the calculation of

$R_0$.

$$\frac{C\Delta I(t) + C\Delta R(t) + C\Delta D(t)}{C\Delta R(t) + C\Delta D(t)} = \frac{\alpha}{\beta + \gamma} = R_0 \tag{10}$$

Here, we used Eq (10) to estimate $R_0$ in order to reduce the noise included in the differences. Note that the above expression is a valid approximation only at the beginning of the spread of the disease.

Thus, based on the above, a coarse estimation of $R_0$ and its corresponding confidence intervals can be provided by solving a linear regression problem using least-squares problem as:

$$\hat{R}_0 = ([\mathbf{C}\Delta\mathbf{R}(t) + \mathbf{C}\Delta\mathbf{D}(t)]^T[\mathbf{C}\Delta\mathbf{R}(t) + \mathbf{C}\Delta\mathbf{D}(t)])^{-1}$$
$$[\mathbf{C}\Delta\mathbf{R}(t) + \mathbf{C}\Delta\mathbf{D}(t)]^T[\mathbf{C}\Delta\mathbf{I}(t) + \mathbf{C}\Delta\mathbf{R}(t) + \mathbf{C}\Delta\mathbf{D}(t)], \tag{11}$$

## Estimation of the case fatality and case recovery ratios for the period January 11-February 10

Here, we denote by $\hat{\gamma}$ the case fatality and by $\hat{\beta}$ the case recovery ratios. Several approaches have been proposed for the calculation of the case fatality ratio (see for example the formula used by the National Health Commission (NHC) of the People's Republic of China [18] for estimating the mortality ratio for the COVID-19 and also the discussion in [19]). Here, we adopt the one used also by the NHC which defines the case mortality ratio as the proportion of the total cases of infected cases, that die from the disease.

Thus, a coarse estimation of the case fatality and recovery ratios for the period under study can be calculated using the reported cumulative infected, recovered and dead cases, by solving a linear regression problem, which for the case fatality ratio reads:

$$\hat{\gamma} = [\mathbf{C}\Delta\mathbf{I}(t)^T\mathbf{C}\Delta\mathbf{I}(t)]^{-1}$$
$$\mathbf{C}\Delta\mathbf{I}(t)^T\mathbf{C}\Delta\mathbf{D}(t), \tag{12}$$

Accordingly, in an analogy to the above, the case recovery ratio reads:

$$\hat{\beta} = [\mathbf{C}\Delta\mathbf{I}(t)^T\mathbf{C}\Delta\mathbf{I}(t)]^{-1}$$
$$\mathbf{C}\Delta\mathbf{I}(t)^T\mathbf{C}\Delta\mathbf{R}(t), \tag{13}$$

As the reported data are just a subset of the actual number of infected and recovered cases including the asymptomatic and/or mild ones, we have repeated the above calculations considering twenty times the reported number of infected and forty times the reported number of recovered in the toal population, while leaving the reported number of dead the same given that their cataloguing is close to the actual number of deaths due to COVID-19.

## Estimation of the "effective" SIRD model parameters

Here we note that the new cases of recovered and deaths at each time time $t$ appear with a time delay with respect to the actual number of infected cases. This time delay is generally unknown but an estimate can be given by clinical studies. However, one could also attempt to provide a coarse estimation of these parameters based only on the reported data by considering the first period of the outbreak and in particular the period from the 11th of January to the 16th of January where the number of infected cases appear to be constant. Thus, based

on Eqs (3) and (4), and the above assumption, the "effective" per day recovery rate $\beta$ and the "effective" per day mortality rate $\gamma$ were computed by solving the least squares problems (see Eqs (2) and (4):

$$
\begin{aligned}
\gamma = [(\mathbf{C}\Delta\mathbf{I}(t-1) &- \mathbf{C}\Delta\mathbf{D}(t-1) - \mathbf{C}\Delta\mathbf{R}(t-1))^T \\
&(\mathbf{C}\Delta\mathbf{I}(t-1) - \mathbf{C}\Delta\mathbf{D}(t-1) - \mathbf{C}\Delta\mathbf{R}(t-1))]^{-1} \\
&(\mathbf{C}\Delta\mathbf{I}(t-1) - \mathbf{C}\Delta\mathbf{D}(t-1) - \mathbf{C}\Delta\mathbf{R}(t-1))^T \Delta\mathbf{D}(t),
\end{aligned}
\tag{14}
$$

and

$$
\begin{aligned}
\beta = [(\mathbf{C}\Delta\mathbf{I}(t-1) &- \mathbf{C}\Delta\mathbf{D}(t-1) - \mathbf{C}\Delta\mathbf{R}(t-1))^T \\
&(\mathbf{C}\Delta\mathbf{I}(t-1) - \mathbf{C}\Delta\mathbf{D}(t-1) - \mathbf{C}\Delta\mathbf{R}(t-1))]^{-1} \\
&(\mathbf{C}\Delta\mathbf{I}(t-1) - \mathbf{C}\Delta\mathbf{D}(t-1) - \mathbf{C}\Delta\mathbf{R}(t-1))^T \Delta\mathbf{R}(t),
\end{aligned}
\tag{15}
$$

As noted, these values do not correspond to the actual per day mortality and recovery rates as these would demand the exact knowledge of the corresponding time delays. Having provided an estimation of the above "effective" approximate values of the parameters $\beta$ and $\gamma$, an approximation of the "effective" infected rate $\alpha$, that is not biased by the assumption of $S = N$, can be obtained by using the SIRD simulator. In particular, in the SIRD model, the values of the $\beta$ and $\gamma$ parameters were set equal to the ones found using the reported data solving the corresponding least squares problems given by Eqs (14) and (15). As initial conditions we have set one infected person on the 16th of November and ran the simulator until the last date for which there are available data (here up to the 10th of February). Then, the optimal value of the infection rate $\alpha$ that fits the reported data was found by "wrapping" around the SIRD simulator an optimization algorithm (such as a nonlinear least-squares solver) to solve the problem:

$$
\underset{\alpha}{\operatorname{argmin}}\Big\{\sum_{t=1}^{M}(w_1 f_t(\alpha;\beta,\gamma)^2 + w_2 g_t(\alpha;\beta,\gamma)^2 + w_3 h_t(\alpha;\beta,\gamma)^2)\Big\},
\tag{16}
$$

where

$$
\begin{aligned}
f_t(\alpha;\beta,\gamma) &= C\Delta I^{SIRD}(t) - C\Delta I(t), \\
g_t(\alpha;\beta,\gamma) &= C\Delta R^{SIRD}(t) - C\Delta R(t), \\
h_t(\alpha;\beta,\gamma) &= C\Delta D^{SIRD}(t) - C\Delta D(t)
\end{aligned}
$$

where, $C\Delta X^{SIRD}(t)$, $(X = I, R, D)$ are the cumulative cases resulting from the SIRD simulator at time $t$; $w_1, w_2, w_3$ correspond to scalars serving in the general case as weights to the relevant functions. For the solution of the above optimization problem we used the function "lsqnonlin" of matlab [20] using the Levenberg-Marquard algorithm.

## Results

As discussed, we have derived results using two different scenarios (see in Methodology). For each scenario, we first present the results for the basic reproduction number as well as the case fatality and case recovery ratios as obtained by solving the least squares problem using a rolling window of an one-day step. For their computation, we used the first six days i.e. from the 11th up to the 16th of January to provide the very first estimations. We then proceeded with the calculations by adding one day in the rolling window as described in the methodology until the 10th of February. We also report the corresponding 90% confidence intervals instead of the more standard 95% because of the small size of the data. For each window, we also report the

corresponding coefficients of determination ($R^2$) representing the proportion of the variance in the dependent variable that is predictable from the independent variables, and the root mean square of error (RMSE). The estimation of $R_0$ was based on the data until January 20, in order to satisfy as much as possible the hypothesis underlying its calculation by Eq (9).

Then, as described above, we provide coarse estimations of the "effective" per day recovery and mortality rates of the SIRD model based on the reported data by solving the corresponding least squares problems. Then, an estimation of the infection rate $\alpha$ was obtained by "wrapping" around the SIRD simulator an optimization algorithm as described in the previous section. Finally, we provide tentative forecasts for the evolution of the outbreak based on both scenarios until the end of February.

### Scenario I: Results obtained using the exact numbers of the reported confirmed cases

Fig 1 depicts an estimation of $R_0$ for the period January 16-January 20. Using the first six days from the 11th of January, $\hat{R}_0$ results in $\sim$ 4.80 (90% CI: 3.36-6.67); using the data until January 17, $\hat{R}_0$ results in $\sim$ 4.60 (90% CI: 3.56-5.65); using the data until January 18, $\hat{R}_0$ results in $\sim$ 5.14 (90%CI: 4.25-6.03); using the data until January 19, $\hat{R}_0$ results in $\sim$ 6.09 (90% CI: 5.02-7.16); and using the data until January 20, $\hat{R}_0$ results in $\sim$ 7.09 (90% CI: 5.84-8.35).

Fig 2 depicts the estimated values of the case fatality ($\hat{\gamma}$) and case recovery ($\hat{\beta}$) ratios for the period January 16 to February 10. The confidence intervals are also depicted with dashed lines. Note that the large variation in the estimated values of $\hat{\beta}$ and $\hat{\gamma}$ should be attributed to the small size of the data and data uncertainty. This is also reflected in the corresponding

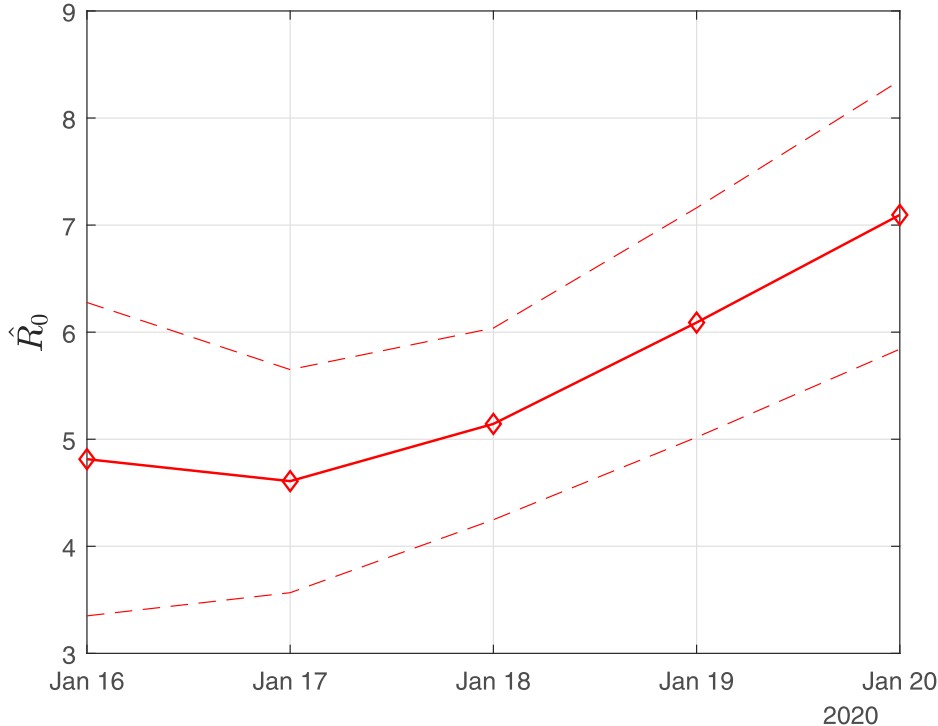

**Fig 1. Scenario I. Estimated values of the basic reproduction number ($R_0$) as computed by least squares using a rolling window with initial date the 11th of January.** The solid line corresponds to the mean value and dashed lines to lower and upper 90% confidence intervals.

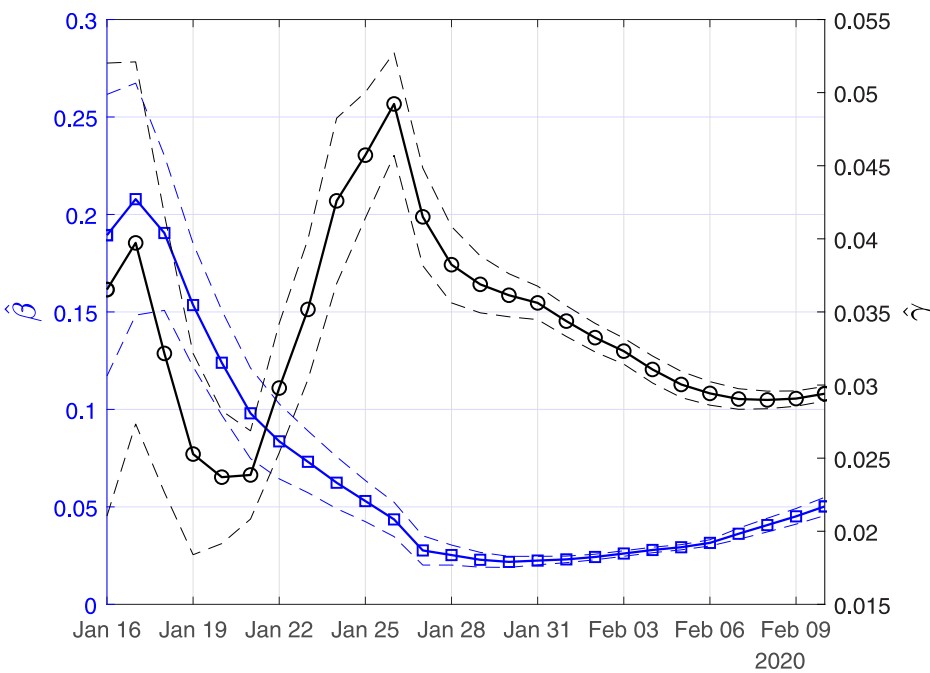

**Fig 2. Scenario I. Estimated values of the case fatality ($\hat{\gamma}$) and case recovery ratios ($\hat{\beta}$) as computed by least squares using a rolling window.** Solid lines correspond to the mean values and dashed lines to lower and upper 90% confidence intervals.

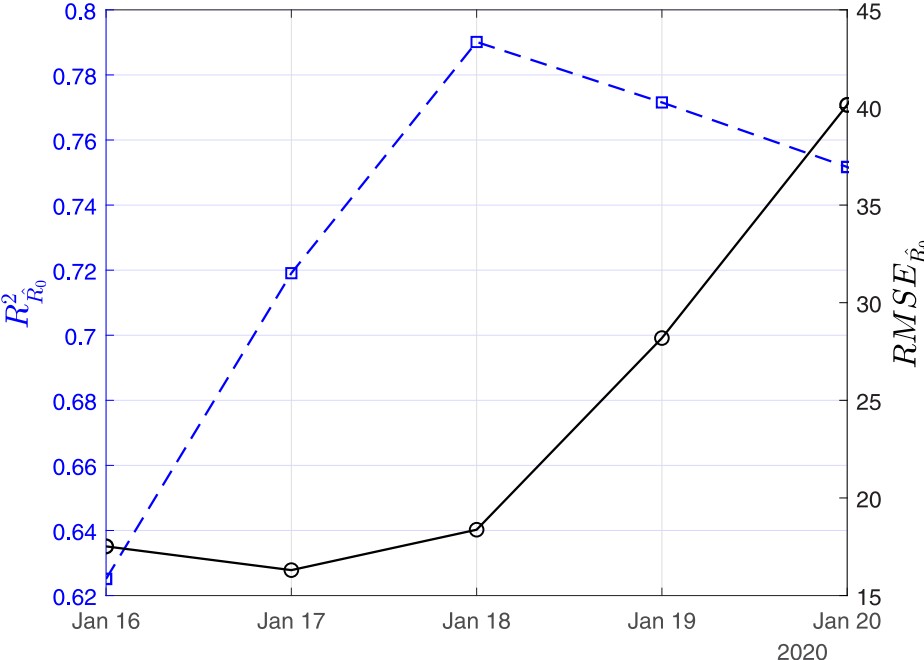

**Fig 3. Scenario I. Coefficient of determination ($R^2$) and root mean square error ($RMSE$) resulting from the solution of the linear regression problem with least-squares for the basic reproduction number ($R_0$).**

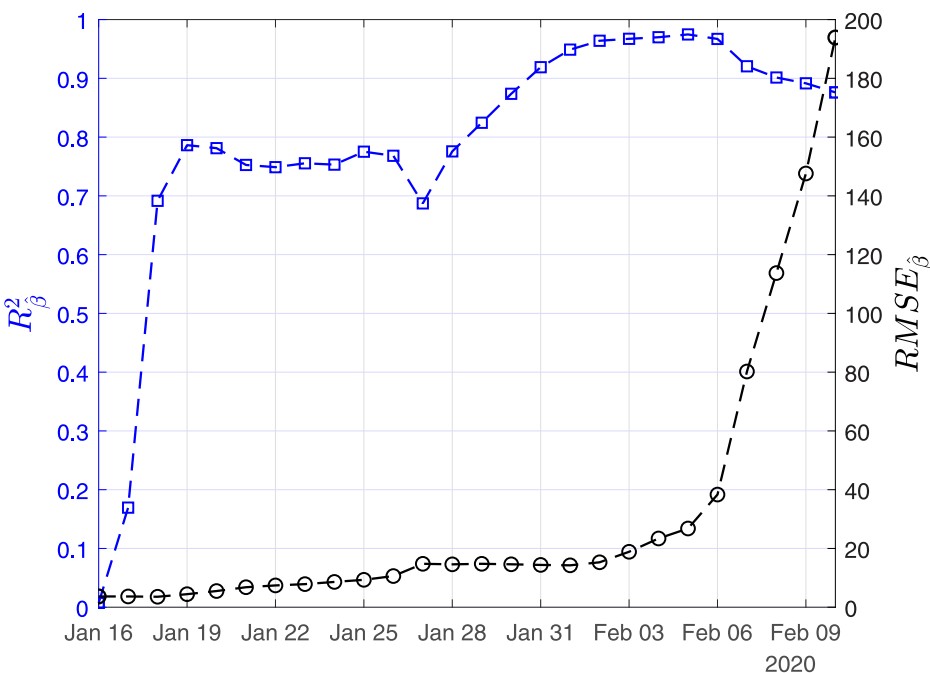

**Fig 4. Scenario I. Coefficient of determination ($R^2$) and root mean square error ($RMSE$) resulting from the solution of the linear regression problem with least-squares for the case recovery ratio ($\hat{\beta}$).**

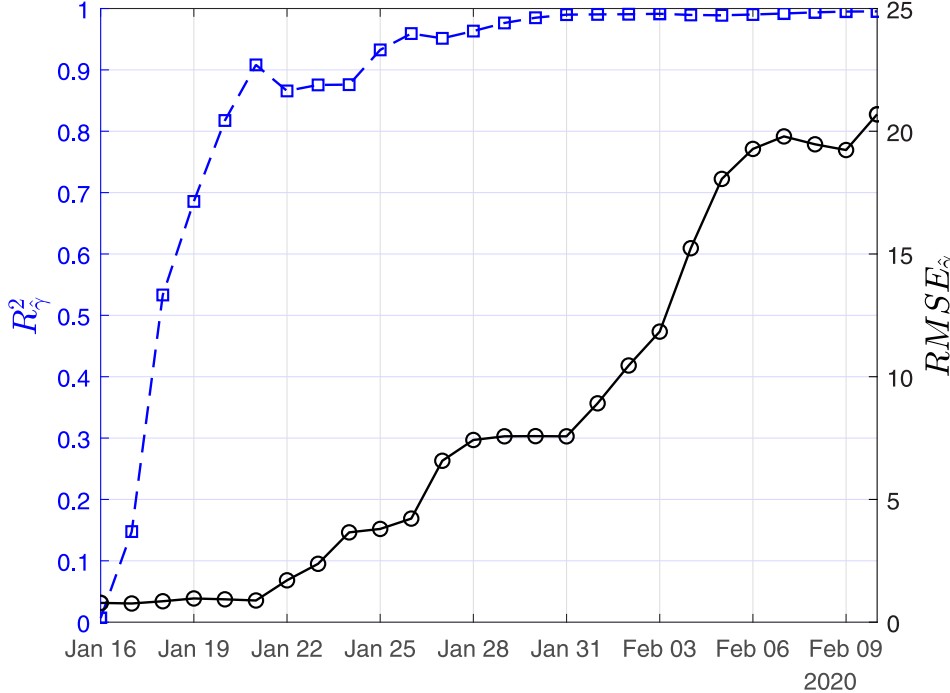

**Fig 5. Scenario I. Coefficient of determination ($R^2$) and root mean square error (RMSE) resulting from the solution of the linear regression problem with least-squares for the case fatality ratio ($\hat{\gamma}$).**

confidence intervals. As more data are taken into account, this variation is significantly reduced. Thus, using all the available data from the 11th of January until the 10th of February, the estimated value of the case fatality ratio $\hat{\gamma}$ is $\sim 2.94\%$ (90% CI: 2.9%-3%) and that of the case recovery ratio $\hat{\beta}$ is $\sim 0.05$ (90% CI: 0.046-0.055). It is interesting to note that as the available data become more, the estimated case recovery ratio increases significantly from the 31th of January (see Fig 2).

In Figs 3, 4 and 5, we show the coefficients of determination ($R^2$) and the root of mean squared errors ($RMSE$) for $\hat{R}_0$, $\hat{\beta}$ and $\hat{\gamma}$, respectively.

The computed approximate values of the "effective" per day mortality and recovery rates of the SIRD model were $\gamma \sim 0.01$ and $\beta \sim 0.064$ (corresponding to a recovery period of $\sim 15$ d). Note that because of the extremely small number of the data used, the confidence intervals have been disregarded. Instead, for our calculations, we have considered intervals of 20% around the expected least squares solutions. Hence, for $\gamma$, we have taken the interval (0.008 and 0.012) and for $\beta$, we have taken the interval between (0.05 and 0.077) corresponding to recovery periods from 13 to 20 days. As described in the methodology, we have also used the SIRD simulator to provide an estimation of the "effective" infection rate $\alpha$ by optimization with $w_1 = 1$, $w_2 = 2$, $w_3 = 2$. Thus, we performed the simulations by setting $\beta = 0.064$ and $\gamma = 0.01$, and as initial conditions one infected, zero recovered and zero dead on November

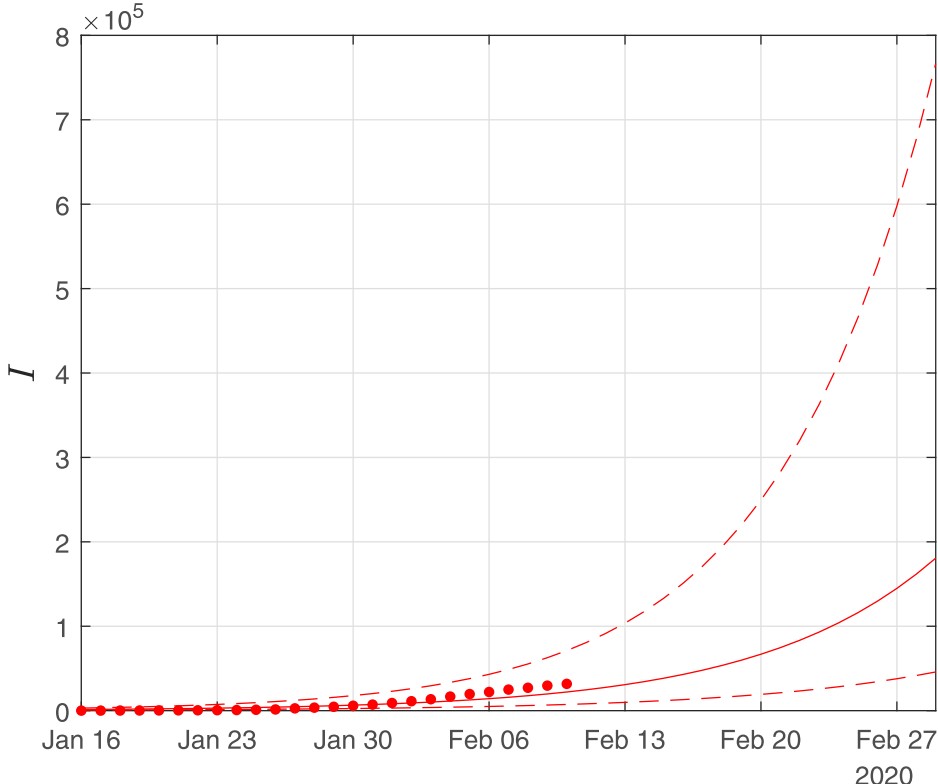

**Fig 6. Scenario I. Simulations until the 29th of February of the cumulative number of infected as obtained using the SIRD model.** Dots correspond to the number of confirmed cases from the 16th of January to the 10th of February. The initial date of the simulations was the 16th of November with one infected, zero recovered and zero deaths. Solid lines correspond to the dynamics obtained using the estimated expected values of the epidemiological parameters $\alpha = 0.191$, $\beta = 0.064d^{-1}$, $\gamma = 0.01$; dashed lines correspond to the lower and upper bounds derived by performing simulations on the limits of the confidence intervals of the parameters.

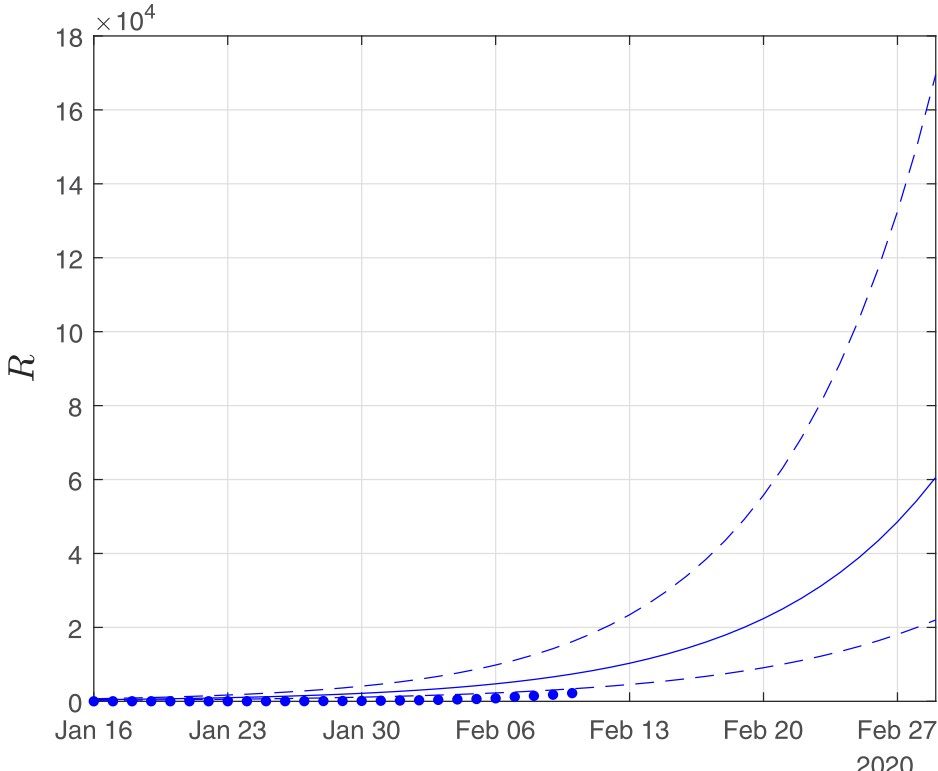

**Fig 7. Scenario I. Simulations until the 29th of February of the cumulative number of recovered as obtained using the SIRD model.** Dots correspond to the number of confirmed cases from the 16th of January to the 10th of February. The initial date of the simulations was the 16th of November with one infected, zero recovered and zero deaths. Solid lines correspond to the dynamics obtained using the estimated expected values of the epidemiological parameters $\alpha = 0.191$, $\beta = 0.064 d^{-1}$, $\gamma = 0.01$; dashed lines correspond to the lower and upper bounds derived by performing simulations on the limits of the confidence intervals of the parameters.

16th 2019, and ran until the 10th of February. The optimal, with respect to the reported confirmed cases from the 11th of January to the 10th of February, value of the infected rate ($\alpha$) was $\sim 0.191$ (90% CI: 0.19-0.192). This corresponds to a mean value of the basic reproduction number $\hat{R}_0 \approx 2.6$. Note that this value is lower compared to the value that was estimated using solely the reported data.

Finally, using the derived values of the parameters $\alpha$, $\beta$, $\gamma$, we performed simulations until the end of February. The results of the simulations are given in Figs 6, 7 and 8. Solid lines depict the evolution, when using the expected (mean) estimations and dashed lines illustrate the corresponding lower and upper bounds as computed at the limits of the confidence intervals of the estimated parameters.

As Figs 6 and 7 suggest, the forecast of the outbreak at the end of February, through the SIRD model is characterized by high uncertainty. In particular, simulations result in an expected number of $\sim 180,000$ infected cases but with a high variation: the lower bound is at $\sim 45,000$ infected cases while the upper bound is at $\sim 760,000$ cases. Similarly for the recovered population, simulations result in an expected number of $\sim 60,000$, while the lower and upper bounds are at $\sim 22,000$ and $\sim 170,000$, respectively. Finally, regarding the deaths, simulations result in an average number of $\sim 9,000$, with lower and upper bounds, $\sim 2,700$ and $\sim 34,000$, respectively.

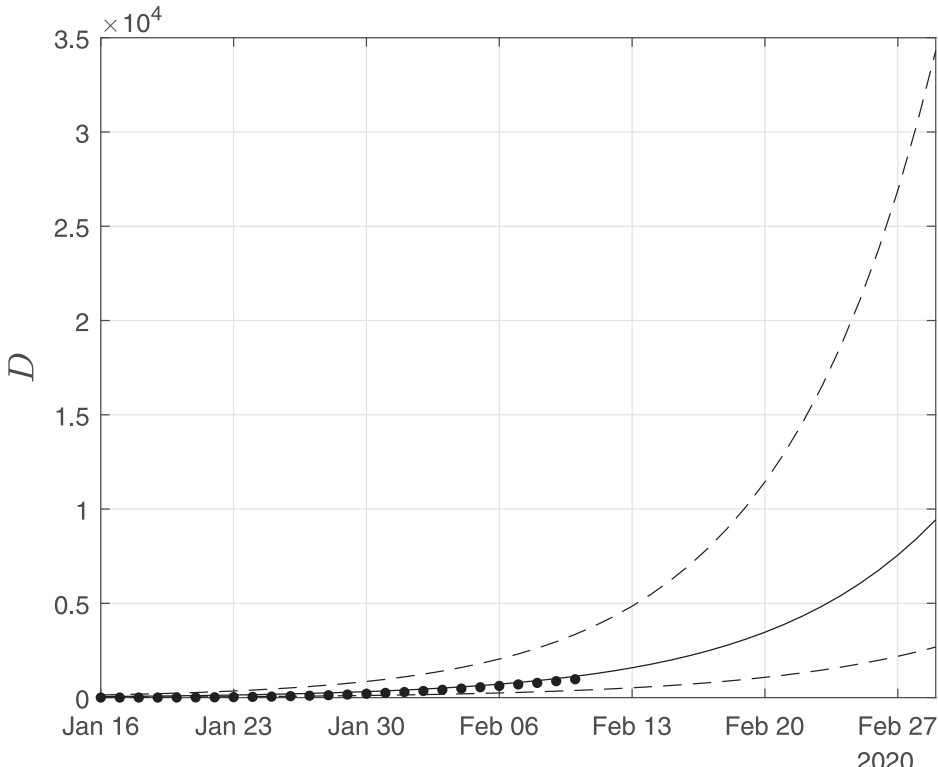

**Fig 8. Scenario I. Simulations until the 29th of February of the cumulative number of deaths as obtained using the SIRD model.** Dots correspond to the number of confirmed cases from 16th of January to the 10th of February. The initial date of the simulations was the 16th of November with one infected, zero recovered and zero deaths. Solid lines correspond to the dynamics obtained using the estimated expected values of the epidemiological parameters $\alpha = 0.191$, $\beta = 0.064d^{-1}$, $\gamma = 0.01$; dashed lines correspond to the lower and upper bounds derived by performing simulations on the limits of the confidence intervals of the parameters.

Thus, the expected trends of the simulations suggest that the mortality rate is lower than the estimated with the current data and thus the death toll is expected to be significantly less compared with the expected trends of the predictions.

As this paper was revised, the reported number of deaths on the 22th February was 2,344, while the expected number of the forecast was ∼4300 with a lower bound of ∼1,300. Regarding the number of infected and recovered cases by February 20, the cumulative numbers of confirmed reported cases were 64,084 infected and 15,299 recovered, while the expected trends of the forecasts were ∼83,000 for the infected and ∼28,000 for the recovered cases. Hence, based on this estimation, the evolution of the epidemic was well within the bounds of our forecasting.

## Scenario II. Results obtained based by taking twenty times the number of infected and forty times the number of recovered people with respect to the confirmed cases

For our illustrations, we assumed that the number of infected is twenty times the number of the confirmed infected and forty times the number of the confirmed recovered people. Based on this scenario, Fig 9 depicts an estimation of $R_0$ for the period January 16-January 20. Using the first six days from the 11th of January to the 16th of January, $\hat{R}_0$ results in 3.2 (90% CI: 2.4-

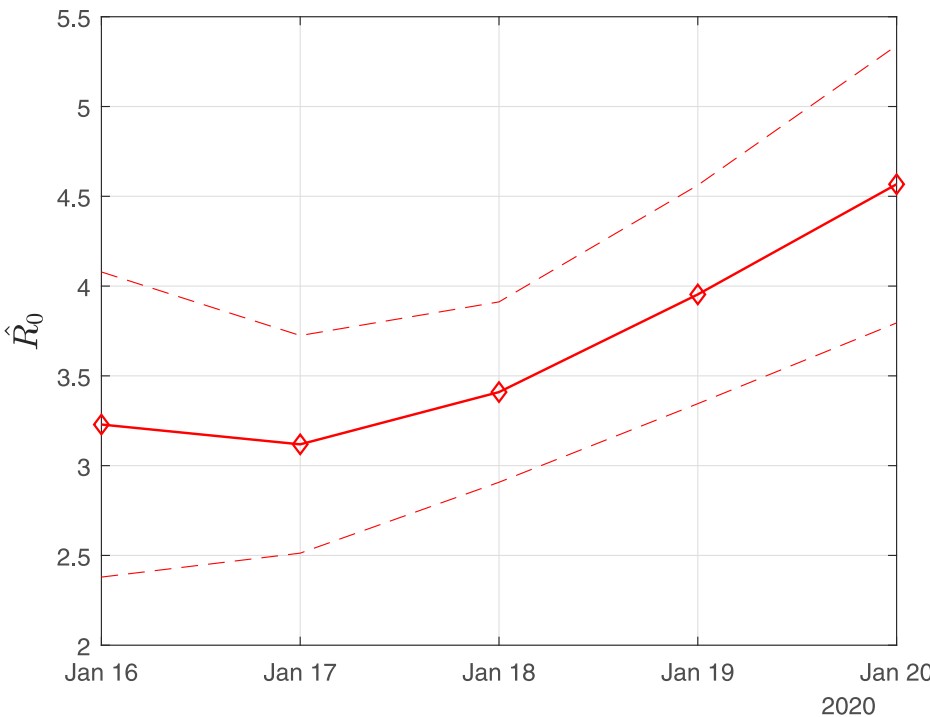

**Fig 9. Scenario II. Estimated values of the basic reproduction number ($R_0$) as computed by least squares using a rolling window with initial date the 11th of January.** The solid line corresponds to the mean value and dashed lines to lower and upper 90% confidence intervals.

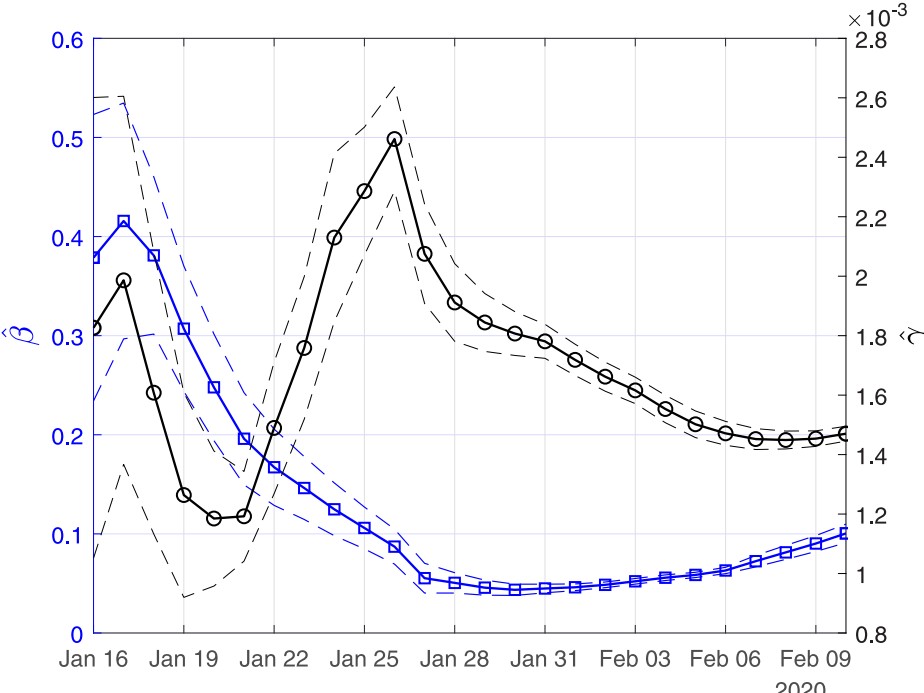

**Fig 10. Scenario II. Estimated values of case fatality ($\hat{\gamma}$) and case recovery ($\hat{\beta}$) ratios, as computed by least squares using a rolling window (see in Methodology).** Solid lines correspond to the mean values and dashed lines to lower and upper 90% confidence intervals.

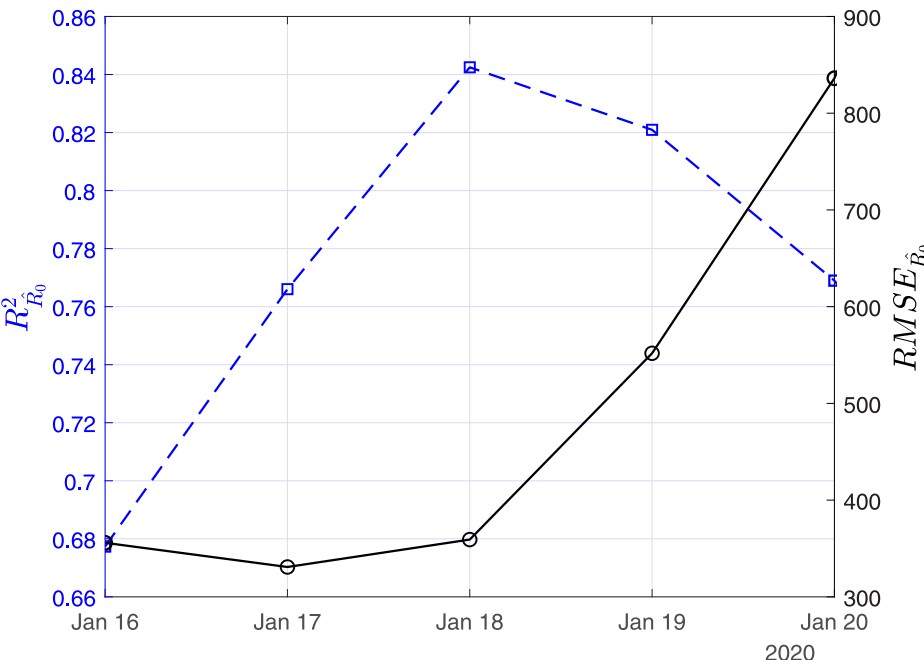

**Fig 11. Scenario II. Coefficient of determination ($R^2$) and root mean square error ($RMSE$) resulting from the solution of the linear regression problem with least-squares for the basic reproduction number ($R_0$).**

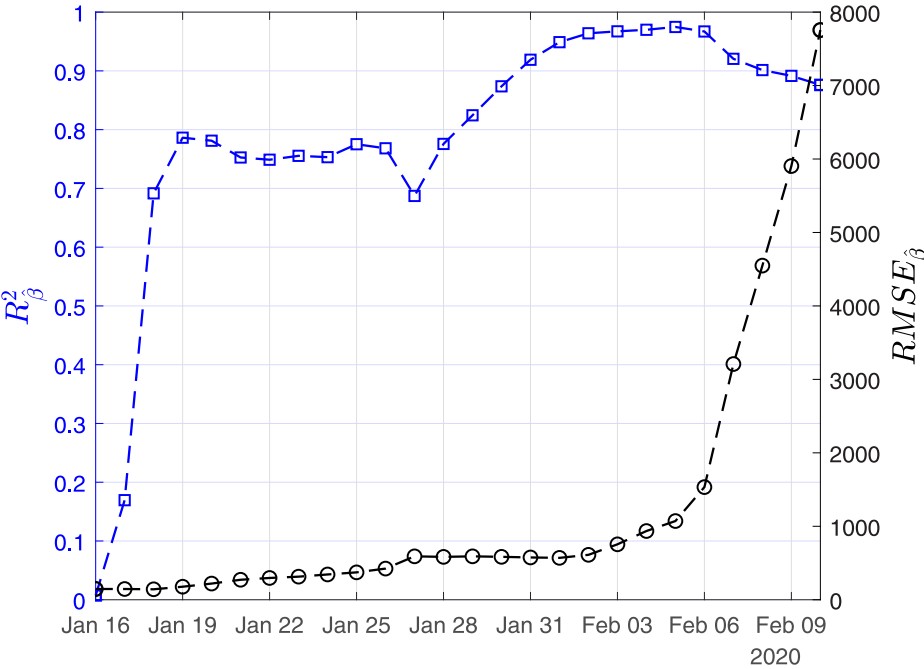

**Fig 12. Scenario II. Coefficient of determination ($R^2$) and root mean square error ($RMSE$) resulting from the solution of the linear regression problem with least-squares for the recovery rate ($\hat{\beta}$).**

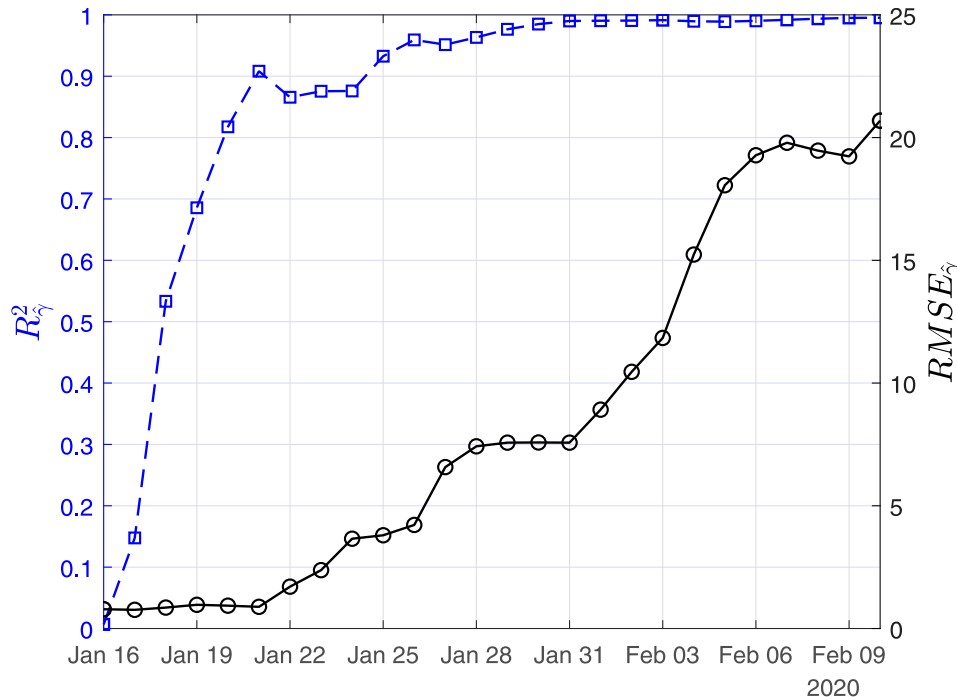

**Fig 13. Scenario II. Coefficient of determination ($R^2$) and root mean square error ($RMSE$) resulting from the solution of the linear regression problem with least-squares for the mortality rate ($\hat{\gamma}$).**

4.0); using the data until January 17, $\hat{R}_0$ results in 3.1 (90% CI: 2.5-3.7); using the data until January 18, $\hat{R}_0$ results in 3.4 (90% CI: 2.9-3.9); using the data until January 19, $\hat{R}_0$ results in 3.9 (90% CI: 3.3-4.5) and using the data until January 20, $\hat{R}_0$ results in 4.5 (90% CI: 3.8-5.3).

It is interesting to note that the above estimation of $R_0$ is close enough to the one reported in other studies (see in the Introduction for a review).

Fig 10 depicts the estimated values of the case fatality ($\hat{\gamma}$) and case recovery ($\hat{\beta}$ ratios for the period January 16 to February 10. The confidence intervals are also depicted with dashed lines. Note that the large variation in the estimated values of $\hat{\beta}$ and $\hat{\gamma}$ should be attributed to the small size of the data and data uncertainty. This is also reflected in the corresponding confidence intervals. As more data are taken into account, this variation is significantly reduced. Thus,using all the (scaled) data from the 11th of January until the 10th of February, the estimated value of the case fatality ratio $\hat{\gamma}$ now drops to $\sim 0.147\%$ (90% CI: 0.144%-0.15%) while that of the case recovery ratio is $\sim 0.1$ (90% CI: 0.091-0.11). It is interesting also to note that as the available data become more, the estimated case recovery ratio increases slightly (see Fig 10), while the case fatality ratio (in the total population) seems to be stabilized at a rate of $\sim 0.15\%$.

In Figs 11, 12 and 13, we show the coefficients of determination ($R^2$) and the root of mean squared errors ($RMSE$), for $\hat{R}_0$, $\hat{\beta}$ and $\hat{\gamma}$, respectively.

The computed values of the "effective" per day mortality and recovery rates of the SIRD model were $\gamma \sim 0.0005$ and $\beta \sim 0.16 d^{-1}$ (corresponding to a recovery period of $\sim 6$ d). Note that because of the extremely small number of the data used, the confidence intervals have been disregarded. Instead, for calculating the corresponding lower and upper bounds in our simulations, we have taken intervals of 20% around the expected least squares solutions.

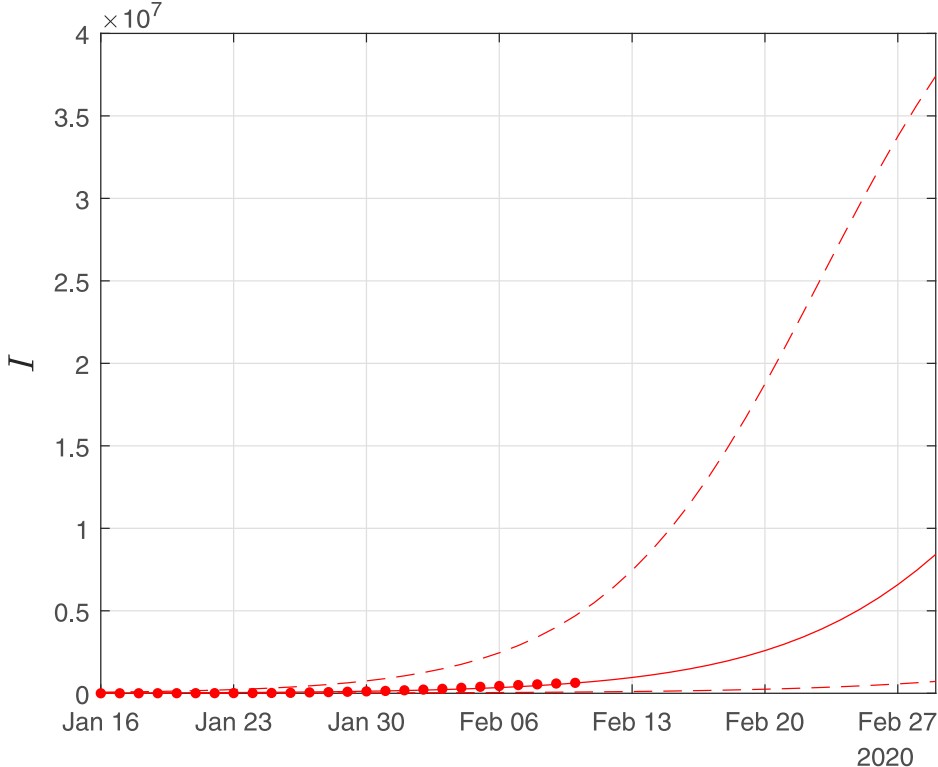

**Fig 14. Scenario II. Simulations until the 29th of February of the cumulative number of infected as obtained using the SIRD model.** Dots correspond to the number of confirmed cases from 16th of Jan to the 10th of February. The initial date of the simulations was the 16th of November with one infected, zero recovered and zero deaths. Solid lines correspond to the dynamics obtained using the estimated expected values of the epidemiological parameters $\alpha = 0.319$, $\beta = 0.16d^{-1}$, $\gamma = 0.0005$; dashed lines correspond to the lower and upper bounds derived by performing simulations on the limits of the confidence intervals of the parameters.

Hence, for $\gamma$ we have taken the interval (0.0004 and 0.0006) and for $\beta$, we have taken the interval between (0.13 and 0.19) corresponding to an interval of recovery periods from 5 to 8 days.

Again, we used the SIRD simulator to provide estimation of the infection rate by optimization setting $w_1 = 1$, $w_2 = 400$, $w_3 = 1$ to balance the residuals of deaths with the scaled numbers of the infected and recovered cases. Thus, to find the optimal infection transmission rate, we used the SIRD simulations with $\beta = 0.16d^{-1}$, and $\gamma = 0.0005$ and as initial conditions one infected, zero recovered, zero deaths on November 16th 2019, and ran until the 10th of February.

The optimal, with respect to the reported confirmed cases from the 11th of January to the 10th of February value of the infected rate ($\alpha$) was found to be $\sim 0.319(90\%$ CI: 0.318-0.32). This corresponds to a mean value of the basic reproduction number $\hat{R}_0 \approx 2$.

Finally, using the derived values of the parameters $\alpha$, $\beta$, $\gamma$, we have run the SIRD simulator until the end of February. The simulation results are given in Figs 14, 15 and 16. Solid lines depict the evolution, when using the expected (mean) estimations and dashed lines illustrate the corresponding lower and upper bounds as computed at the limits of the confidence intervals of the estimated parameters.

Again as Figs 15 and 16 suggest, the forecast of the outbreak at the end of February, through the SIRD model is characterized by high uncertainty. In particular, in Scenario II, by February

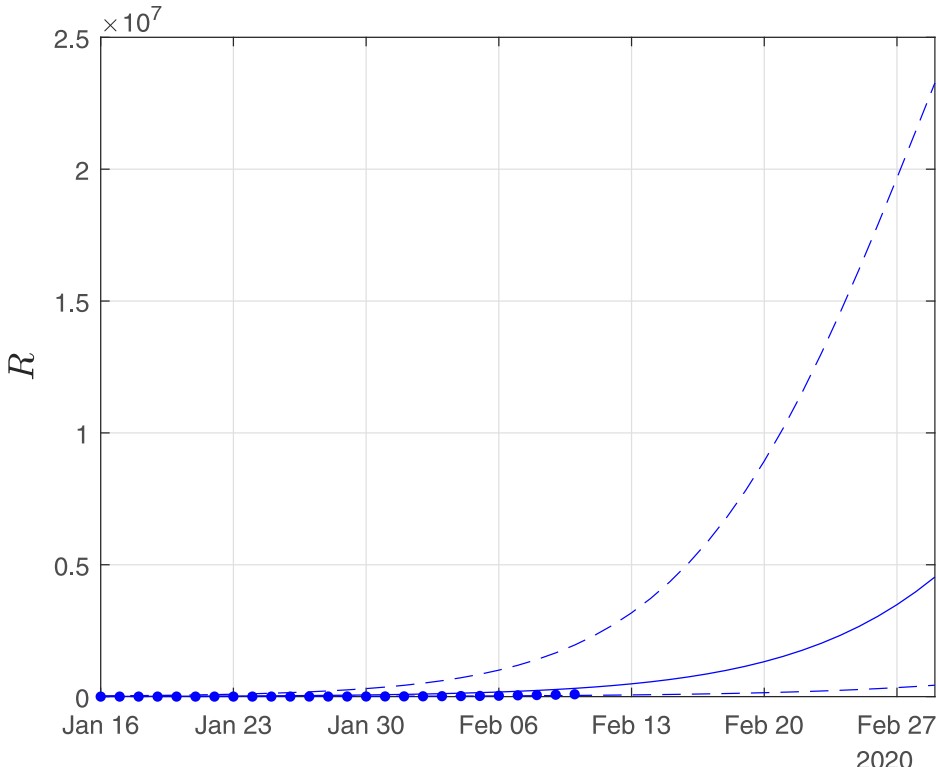

**Fig 15. Scenario II. Simulations until the 29th of February of the cumulative number of recovered as obtained using the SIRD model.** Dots correspond to the number of confirmed cases from 16th of January to the 10th of February. The initial date of the simulations was the 16th of November, with one infected, zero recovered and zero deaths. Solid lines correspond to the dynamics obtained using the estimated expected values of the epidemiological parameters $\alpha = 0.319$, $\beta = 0.16 d^{-1}$, $\gamma = 0.0005$; dashed lines correspond to the lower and upper bounds derived by performing simulations on the limits of the confidence intervals of the parameters.

29, simulations result in an expected actual number of ∼8m infected cases (corresponding to a ∼13% of the total population) with a lower bound at ∼720,000 and an upper bound at ∼37m cases. Similarly, for the recovered population, simulations result in an expected actual number of ∼4.5m (corresponding to a 8% of the total population), while the lower and upper bounds are at ∼430,000 and ∼23m, respectively. Finally, regarding the deaths, simulations under this scenario result in an average number of ∼14,000, with lower and upper bounds at ∼900 and ∼100,000.

Importantly, under this scenario, the simulations shown in Fig 14 suggest a decline of the outbreak at the end of February. Table 1 summarizes the above results for both scenarios.

We note that the results derived under Scenario II seem to predict a slowdown of the outbreak in Hubei after the end of February.

## Discussion

We have proposed a methodology for the estimation of the key epidemiological parameters as well as the modelling and forecasting of the spread of the COVID-19 epidemic in Hubei, China by considering publicly available data from the 11th of January 2020 to the 10th of February 2020.

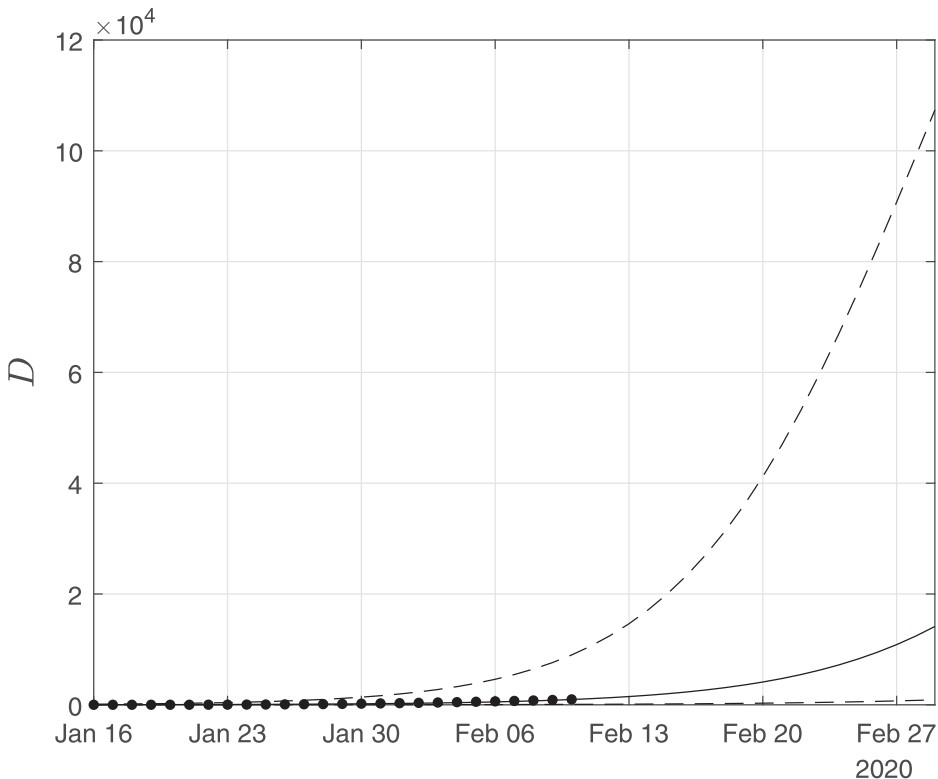

**Fig 16. Scenario II. Simulations until the 29th of February of the cumulative number of deaths as obtained using the SIRD model.** Dots correspond to the number of confirmed cases from the 16th of November to the 10th of February. The initial date of the simulations was the 16th of November with zero infected, zero recovered and zero deaths. Solid lines correspond to the dynamics obtained using the estimated expected values of the epidemiological parameters $\alpha = 0.319$, $\beta = 0.16 d^{-1}$, $\gamma = 0.0005$; dashed lines correspond to the lower and upper bounds derived by performing simulations on the limits of the confidence intervals of the parameters.

By the time of the acceptance of our paper, according to the official data released on the 29th of February, the cumulative number of confirmed infected cases in Hubei was ∼67,000, that of recovered was ∼31,300 and the death toll was ∼2,800. These numbers are within the lower bounds and expected trends of our forecasts from the 10th of February that are based on Scenario I. Importantly, by assuming a 20-fold scaling of the confirmed cumulative number of the infected cases and a 40-fold scaling of the confirmed number of the recovered cases in the total population, forecasts show a decline of the outbreak in Hubei at the end of February. Based on this scenario the case fatality rate in the total population is of the order of ∼0.15%.

At this point we should note that our SIRD modelling approach did not take into account many factors that play an important role in the dynamics of the disease such as the effect of the incubation period in the transmission dynamics, the heterogeneous contact transmission network, the effect of the measures already taken to combat the epidemic, the characteristics of the population (e.g. the effect of the age, people who had already health problems). Also the estimation of the model parameters is based on an assumption, considering just the first period in which the first cases were confirmed and reported. Of note, COVID-19, which is thought to be principally transmitted from person to person by respiratory droplets and fomites without excluding the possibility of the fecal-oral route [21] had been spreading for at least over a month and a half before the imposed lockdown and quarantine of Wuhan on January 23, having thus infected unknown numbers of people. The number of asymptomatic and mild cases

**Table 1. Model parameters, their computed values and forecasts for the Hubei province under two scenarios: (I) using the exact values of confirmed cases or (II) using estimations for infected and recovered (twenty and forty times the number of confirmed cases, respectively).**

| Estimations | Symbol | Parameter | Computed values | 90% CI |
|---|---|---|---|---|
| **Scenario I: Exact numbers for confirmed cases** | | | | |
| Based on linear regression of the data | $R_0$ | Basic reproduction number | | |
| | | 11-16 Jan | 4.80 | 3.36-6.67 |
| | | 11-17 Jan | 4.60 | 3.56-5.65 |
| | | 11-18 Jan | 5.14 | 4.25-6.03 |
| | $\hat{\beta}$ | case recovery ratio | 0.05 | 0.046-0.055 |
| | $\hat{\gamma}$ | case fatality ratio | 2.94% | 2.9%-3% |
| Based on the SIRD simulator (Nov 16-Feb 10) | $R_0$ | Basic reproduction number | 2.6 | - |
| | $\alpha$ | infection rate | 0.191 | 0.19-0.192 |
| Forecast to Feb 29 (Cumulative) | | infected | 180,000 | 45,000-760,000 |
| | | recovered | 60,000 | 22,000-170,000 |
| | | deaths | 9,000 | 2,700-34,000 |
| **Scenario II: x20 Infected, x40 recovered of confirmed cases** | | | | |
| Based on linear regression of the data | $R_0$ | Basic reproduction number | | |
| | | 11-16 Jan | 3.2 | 2.4-4.0 |
| | | 11-17 Jan | 3.1 | 2.5-3.7 |
| | | 11-18 Jan | 3.4 | 2.9-3.9 |
| | $\hat{\beta}$ | case recovery ratio | 0.1 | 0.091-0.11 |
| | $\hat{\gamma}$ | case fatality ratio | 0.147% | 0.144%-0.15% |
| Based on the SIRD simulator (Nov 16-Feb 10) | $R_0$ | Basic reproduction number | 2 | - |
| | $\alpha$ | infection rate | 0.319 | 0.318-0.32 |
| Forecast to Feb 29 (Cumulative) | | infected | 8m | 720,000-37m |
| | | recovered | 4.5m | 430,000-23m |
| | | deaths | 14,000 | 900-100,000 |

with subclinical manifestations that probably did not present to hospitals for treatment may be substantial; these cases, which possibly represent the bulk of the COVID-19 infections, remain unrecognized, especially during the influenza season [22]. This highly likely gross under-detection and underreporting of mild or asymptomatic cases inevitably throws severe disease courses calculations and death rates out of context, distorting epidemiologic reality.

Another important factor that should be taken into consideration pertains to the diagnostic criteria used to determine infection status and confirm cases. A positive PCR test was required to be considered a confirmed case by China's Novel Coronavirus Pneumonia Diagnosis and Treatment program in the early phase of the outbreak [14]. However, the sensitivity of nucleic acid testing for this novel viral pathogen may only be 30-50%, thereby often resulting in false negatives, particularly early in the course of illness. To complicate matters further, the guidance changed in the recently-released fourth edition of the program on February 6 to allow for diagnosis based on clinical presentation, but only in Hubei province [14].

The swiftly growing epidemic seems to be overwhelming even for the highly efficient Chinese logistics that did manage to build two new hospitals in record time to treat infected patients. Supportive care with extracorporeal membrane oxygenation (ECMO) in intensive care units (ICUs) is critical for severe respiratory disease. Large-scale capacities for such level of medical care in Hubei province, or elsewhere in the world for that matter, amidst this public health emergency may prove particularly challenging. We hope that the results of our analysis contribute to the elucidation of critical aspects of this outbreak so as to contain the

novel coronavirus as soon as possible and mitigate its effects regionally, in mainland China, and internationally.

## Conclusion

In the digital and globalized world of today, new data and information on the novel coronavirus and the evolution of the outbreak become available at an unprecedented pace. Still, crucial questions remain unanswered and accurate answers for predicting the dynamics of the outbreak simply cannot be obtained at this stage. We emphatically underline the uncertainty of available official data, particularly pertaining to the true baseline number of infected (cases), that may lead to ambiguous results and inaccurate forecasts by orders of magnitude, as also pointed out by other investigators [1, 17, 22].

## Supporting information

**S1 Table. Reported cumulative numbers of cases for the Hubei region, China for the period January 11-February 10.**
(PDF)

## Author Contributions

**Conceptualization:** Cleo Anastassopoulou, Constantinos Siettos.

**Data curation:** Cleo Anastassopoulou.

**Formal analysis:** Lucia Russo, Constantinos Siettos.

**Investigation:** Athanasios Tsakris.

**Methodology:** Lucia Russo, Constantinos Siettos.

**Writing – original draft:** Cleo Anastassopoulou, Constantinos Siettos.

**Writing – review & editing:** Cleo Anastassopoulou, Athanasios Tsakris, Constantinos Siettos.

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
