## [Decision Letter · Decision Letter 0]

20 Feb 2020

PONE-D-20-04084

Data-Based Analysis, Modelling and Forecasting of the novel Coronavirus (2019-nCoV) outbreak

PLOS ONE

Dear Professor Siettos,

Thank you again for submitting your manuscript to PLOS ONE!

Your manuscript has been reviewed by 2 reviewers, and we invite you to submit a revised version of the manuscript that addresses the points raised during the review process. Please find the reviews copied below.

We would appreciate receiving your revised manuscript by Apr 05 2020 11:59PM. Please include the following items when submitting your revised manuscript:

*** Also, as part of an initiative between PLOS (and other publishers) and the World Health Organisation to ensure that all relevant clinical information about this outbreak is shared quickly, we would like to ask your permission to notify the WHO directly that about your study/manuscript. If you have an existing preprint of your manuscript posted, we can simply notify the WHO of the preprint identifier. Alternatively, we can send them a copy of your manuscript file. More information on this initiative can be found here: https://wellcome.ac.uk/press-release/sharing-research-data-and-findings-relevant-novel-coronavirus-covid-19-outbreak. ***

We look forward to receiving your revised manuscript.

Kind regards,

Artur Arikainen

Associate Editor

PLOS ONE

-------------

--To enhance the reproducibility of your results, we recommend that if applicable you deposit your laboratory protocols in protocols.io, where a protocol can be assigned its own identifier (DOI) such that it can be cited independently in the future. For instructions see: http://journals.plos.org/plosone/s/submission-guidelines#loc-laboratory-protocols

-Please note while forming your response to reviewers that, if your article is accepted, you may have the opportunity to make the peer review history publicly available. The record will include editor decision letters (with reviews) and your responses to reviewer comments. If eligible, we will contact you to opt in or out.

Journal Requirements:

Reviewers' comments:

Reviewer's Responses to Questions

**Comments to the Author**

1. Is the manuscript technically sound, and do the data support the conclusions?

Reviewer #1: Yes

Reviewer #2: Yes

2. Has the statistical analysis been performed appropriately and rigorously? 

Reviewer #1: Yes

Reviewer #2: Yes

3. Have the authors made all data underlying the findings in their manuscript fully available?

Reviewer #1: Yes

Reviewer #2: Yes

4. Is the manuscript presented in an intelligible fashion and written in standard English?

Reviewer #1: Yes

Reviewer #2: Yes

5. Review Comments to the Author

Reviewer #1: I think that this topic is very current and important and that these estimates can contribute to the effort to face the current epidemic in China and the potential pandemic of this new coronavirus. I suggest adapting the title to the new nomenclature of the virus (SARS-CoV-2) or that of the disease (COVID-19).

Reviewer #2: 1. On p. 2, in the Introduction, the new official name of “COVID-19” has been approved by the WHO for this virus since this article was submitted. In the second paragraph of Methodology, R0 should be R_0

2. p. 3, in equation (2), it is unclear why the second term only has $\\alpha S(t-1)I(t-1)$, but does not have N in the denominator, similarly to an identical term in equation (1). Of course, it is very likely just a typo, since an expression for R_0 in equation (5) is correct.

3. When introducing the system (1)-(4), it is worth mentioning that it is defined over an interval $t=1,2,\\ldots$, with the corresponding initial condition $S(0)=N$, $I(0)=1$, $R(0)=D(0)=0$.

4. p. 3, in the first line of subsection 2.1 I suggest to explicitly write \\Delta I(t)=I(t)-I(t-1) etc to make it clear from the start, over which time interval the changes are measured (this is now stated later on the same page).

5. p. 3, in the right-hand side of equation (7), there is an extra multiplier $(t-1)$, and in the same right-hand side, it is worth to explicitly write $N$ in the denominator. Then, after equation (8), it is worth writing $S(t-1)\\approx N$, which would then give expression (9).

6. Bearing in mind that according to (9), $R0$ is related to changes in the infected/recovered/dead individuals over a single time interval, it is not clear how this then translates into expression (10), which contains cumulative time changes over the course of an epidemic. The authors should provide an explanation of how this expression was produced, and also mention in the text what is denoted by a prime in this expression. A similar question applies to expression (11), for which an explanation should also be provided.

7. p. 4, some brief explanation should be provided to explain why the weights in (12) are also square if one is minimising a sum of squares that are already non-negative.

8. p. 4, it is worth mentioning why for statistical analysis a 90% confidence interval was chosen instead of a more standard 95%.

9. In Scenario I, it is not clear why the value of R0 was only estimated using the time interval from 16 January to 20 January, while the recovery and mortality rates were obtained over a much longer interval up to 10 February. Similarly, with very high variability in the values of $\\widehat{\\beta}$ and $\\widehat{\\gamma}$ as shown in Figure 2, some comment should be provided as to how the specific values of $\\beta$ and $\\gamma$ were chosen on p. 2 to obtain an estimate of the mean reproduction number. Also, the authors should comment on how this value corresponds with the observed values as depicted in Fig. 1, or its continuation over a longer time period.

The values of $\\widehat{\\beta}$ and $\\widehat{\\gamma}$ shown in Fig. 2 appear to vary by between 4 and almost 8 times between their minima and maxima. The authors should comment on what such large variation could be attributed to, and what it means for being able to discern their `actual’ values for the purposes of computing the basic reproduction number.

All of these questions also apply to Scenario II.

10. When revising this paper, it is worth having a look at https://www.worldometers.info/coronavirus/

to see how the observed numbers of new cases identified in Hubei province since this paper was submitted, fit with model predictions. This should give another week worth of data points, and the authors could briefly comment on which of their scenarios appears to be closer to the current observed state of an epidemic in Hubei province.

6. PLOS authors have the option to publish the peer review history of their article (what does this mean?). If published, this will include your full peer review and any attached files.

Reviewer #1: Yes: Andre Ricardo Ribas Freitas

Reviewer #2: No

---

## [Author Response · Author response to Decision Letter 0]

23 Feb 2020

Response Letter

We would like to thank both reviewers for the time and effort they put in reviewing our manuscript. We appreciate their rapid responses and their positive and constructive comments.

We would also like to thank the handling Editor for all his efforts to oversight the review process.

Below, we tried to respond point-by-point to the comments of the reviewers and revised our manuscript accordingly. We believe that the manuscript is now ready for publication.

Reviewer #1: I think that this topic is very current and important and that these estimates can contribute to the effort to face the current epidemic in China and the potential pandemic of this new coronavirus. I suggest adapting the title to the new nomenclature of the virus (SARS-CoV-2) or that of the disease (COVID-19).

Response

We thank the reviewer for his/her positive evaluation of our work. We have now updated the title and the text with the new nomenclature of COVID-19.

Reviewer #2: 

1. On p. 2, in the Introduction, the new official name of “COVID-19” has been approved by the WHO for this virus since this article was submitted. In the second paragraph of Methodology, R0 should be R_0

Response

That what also the comment of the first reviewer. We have updated the title and the text with the new name of the novel coronavirus and the disease. We have also corrected the typo.

2. p. 3, in equation (2), it is unclear why the second term only has $\\alpha S(t-1)I(t-1)$, but does not have N in the denominator, similarly to an identical term in equation (1). Of course, it is very likely just a typo, since an expression for R_0 in equation (5) is correct.

Response

Thank you. Indeed, it was just a typo, which has been corrected in the revised manuscript.

3. When introducing the system (1)-(4), it is worth mentioning that it is defined over an interval $t=1,2,\\ldots$, with the corresponding initial condition $S(0)=N$, $I(0)=1$, $R(0)=D(0)=0$.

Response

We have now included the clarification.

4. p. 3, in the first line of subsection 2.1 I suggest to explicitly write \\Delta I(t)=I(t)-I(t-1) etc to make it clear from the start, over which time interval the changes are measured (this is now stated later on the same page).

Response

We have updated the corresponding equalities, as suggested.

5. p. 3, in the right-hand side of equation (7), there is an extra multiplier $(t-1)$, and in the same right-hand side, it is worth to explicitly write $N$ in the denominator. Then, after equation (8), it is worth writing $S(t-1)\\approx N$, which would then give expression (9).

Response

There was a typo there, which we have corrected. Thank you for noticing. We have also written $S(t-1)\\approx N$.

6. Bearing in mind that according to (9), $R0$ is related to changes in the infected/recovered/dead individuals over a single time interval, it is not clear how this then translates into expression (10), which contains cumulative time changes over the course of an epidemic. The authors should provide an explanation of how this expression was produced, and also mention in the text what is denoted by a prime in this expression. A similar question applies to expression (11), for which an explanation should also be provided.

Response

The reviewer is correct to raise this point. One has two options here: either to do the regression using the differences as they appear in (9), or using their cumulative sums. If one sums up both sides of (7) and (8) over time, then one gets their cumulative sums instead of the differences. However, this way one reduces the noise in the regression. 

Below Eq. 9, we have now added the following paragraphs and a new Equation (Eq. 10) to explain this:

“At this point, the regression can be done either by using the differences per se, or by using the corresponding cumulative functions (instead of the differences for the calculation of $R_0$ using Eq.(\\ref{eq9})). Indeed, it is easy to prove that by summing up both sides of Eq.(\\ref{eq7}) and Eq.(\\ref{eq8}) over time and then dividing them we get the following equivalent expression for the calculation of $R_0$.”

Here, we have least squares using Eq. (\\ref{eq9b}) to estimate $R_0$ in order to reduce the noise included in the differences. Note that the above expression is a valid approximation only at the beginning of the spread of the disease.

Thus, based on the above, a coarse estimation of $R_0$ and its corresponding confidence intervals can be provided by solving a linear regression problem using least-squares problem as:

Eq. 10.

We also explain that in the above the prime $'$ is for the transpose operation.

7. p. 4, some brief explanation should be provided to explain why the weights in (12) are also square if one is minimising a sum of squares that are already non-negative.

Response

Indeed, the weights should be out of the parenthesis. 

8. p. 4, it is worth mentioning why for statistical analysis a 90% confidence interval was chosen instead of a more standard 95%.

Response

We now mention explicitly why the 90% CI was used. In the Results section, we have now added the following sentence:

“We also report the corresponding 90% confidence intervals instead of the more standard 95% because of the small size of the data”

9. In Scenario I, it is not clear why the value of R0 was only estimated using the time interval from 16 January to 20 January, while the recovery and mortality rates were obtained over a much longer interval up to 10 February.

Response 

We have taken just the interval from 16th to 20th of January to compute R0 in order to be as close as possible to the hypothesis of S~N. As the epidemic evolved with more cases, this hypothesis is violated. 

On the other hand, the computations of the recovery and mortality rates are getting more robust as more data are introduced. In the Results section, we have now added the following sentence to better explain this point.

“The estimation of $R_0$ was based on the data until January 20 in order to satisfy as much as possible the hypothesis underlying its calculation by Eq.(\\ref{eq9})”

10. Similarly, with very high variability in the values of $\\widehat{\\beta}$ and $\\widehat{\\gamma}$ as shown in Figure 2, some comment should be provided as to how the specific values of $\\beta$ and $\\gamma$ were chosen on p. 2 to obtain an estimate of the mean reproduction number. Also, the authors should comment on how this value corresponds with the observed values as depicted in Fig. 1, or its continuation over a longer time period.

Response 

Exactly, due to this variability, we don’t compute R0 simply by taking the fraction of the computed by regression values of \\beta \\ gamma and \\alpha. Actually, we don’t compute \\alpha by regression. We compute R0 explicitly through Eq. 9 (actually through the corresponding cumulative functions). 

We have now added the following sentence below Eq.9 to make this clear:

“Note that one can use directly Eq.(\\ref{eq9}) to compute $R_0$ with regression, without the need to compute first the other parameters, i.e. $\\beta$, $\\gamma$ and $\\alpha$.”

11. The values of $\\widehat{\\beta}$ and $\\widehat{\\gamma}$ shown in Fig. 2 appear to vary by between 4 and almost 8 times between their minima and maxima. The authors should comment on what such large variation could be attributed to, and what it means for being able to discern their `actual’ values for the purposes of computing the basic reproduction number.

Response

We have now added the following sentence to comment on this in the revised manuscript. 

“Note that the large variation in the estimated values of $\\beta$ and $\\gamma$ may be accounted to the small size of the data and data uncertainty. This is also reflected in the corresponding confidence intervals. As more data are taken into account, this variation is significantly reduced.”

12. All of these questions also apply to Scenario II.

Response

We have now updated the text accordingly also for the second Scenario.

13. When revising this paper, it is worth having a look at

https://www.worldometers.info/coronavirus/

to see how the observed numbers of new cases identified in Hubei province since this paper was submitted, fit with model predictions. This should give another week worth of data points, and the authors could briefly comment on which of their scenarios appears to be closer to the current observed state of an epidemic in Hubei province.

Response

We thank the reviewer for this comment. We have re-done the computations considering that the actual number of infected in the population is 20 times the reported confirmed cases of infected and 40 times the number of confirmed cases of recovered, while keeping the number of deaths unchanged. 

It seems that the forecasts from the simulations based on this Scenario are closer to the scaled current observed data. Thus, we have revised our manuscript as follows:

A. At the end of the Methodology section (before the beginning of subsection 2.2) we have added the following paragraph:

“As the reported data are just a sample of the actual number of infected and recovered cases including the asymptomatic and/or mild ones, we have repeated the above calculations considering twenty times the reported number of infected and forty times the reported number of recovered in the population, while leaving the reported number of deaths the same, given that their cataloguing is close to the actual number of deaths due to COVID-19.”

B. At the end the subsection with the results of Scenario I we have added the following paragraph:

“Furthermore, simulations reveal that the confirmed cumulative number of deaths is significantly smaller than the lower bound of the simulations. This suggests that the mortality rate is considerably lower than the estimated one based on the officially reported data. Thus, it is expected that the actual numbers of the infected, and consequently of the recovered ones too, are considerably larger than reported. Hence, we assessed the dynamics of the epidemic considering a different scenario that we present in the following subsection.”

C. Before the end of the subsection with the results of Scenario II, we have added the following paragraph:

“We note, that the results derived under Scenario II seem to better reflect the actual situation as the reported number of deaths is within the average and lower limits of the SIRD simulations. In particular, as this paper was revised, the reported number of deaths on the 22th February was ~2,346, while the lower bound of the forecast is ~2,900. This indicates an even lower mortality rate than that of ~0.147%, and thus an even larger actual number of infected (and recovered) cases in the population. Regarding the number of infected and recovered cases by February 20, the cumulative numbers of confirmed reported cases were 64,084 infected and 15,299 recovered. Thus, the corresponding scaled numbers are 1,281,680 infected and 611,960 recovered. Based on Scenario II, for the 22th of February, our simulations give a total number of ~758,000 infected with ~1.8m as an upper bound, and a total of ~520,000 recovered with a total of 1.1m as an upper bound.

Hence, based on this estimation of the actual numbers, the evolution of the epidemic is within the bounds of our forecasting.”

---

## [Editor Report · Decision Letter 1]

2 Mar 2020

Data-Based Analysis, Modelling and Forecasting of the COVID-19 outbreak

PONE-D-20-04084R1

Dear Dr. Siettos,

We are pleased to inform you that your manuscript has been judged scientifically suitable for publication and will be formally accepted for publication once it complies with all outstanding technical requirements.

With kind regards,

Sreekumar Othumpangat, PhD

Academic Editor

PLOS ONE

Additional Editor Comments (optional):

This manuscript is well organized and have improved after the incorporation of the reviewers suggestions. This manuscript has utmost importance due to the rapid increase in COVID-19 cases through out the world with the mortality rate reaching 2 %, which is 20 times more than influenza based cases.
---

## [Editor Report · Acceptance letter]

17 Mar 2020

PONE-D-20-04084R1 

Data-Based Analysis, Modelling and Forecasting of the COVID-19 outbreak 

Dear Dr. Siettos:

I am pleased to inform you that your manuscript has been deemed suitable for publication in PLOS ONE. Congratulations! Your manuscript is now with our production department. 

With kind regards,

on behalf of

Dr. Sreekumar Othumpangat 

Academic Editor

PLOS ONE